# Evaluation of Immunogenicity and Safety of Vero Cell-Derived Inactivated COVID-19 Vaccine in Older Patients with Hypertension and Diabetes Mellitus

**DOI:** 10.3390/vaccines10071020

**Published:** 2022-06-25

**Authors:** Yuntao Zhang, Haiping Chen, Jun Lv, Tao Huang, Ruizhi Zhang, Dongjuan Zhang, Linyun Luo, Sheng Wei, Xiaoqin Liu, Shangxiao Zhang, Qiuyue Mu, Rongdong Huang, Jiao Huang, Yanhui Xiao, Yunkai Yang, Yuting Han, Hao Gong, Qinghu Guan, Fangqin Xie, Hui Wang, Liming Li, Xiaoming Yang

**Affiliations:** 1China National Biotech Group Co., Ltd., Beijing 100024, China; zhangyuntao@sinopharm.com (Y.Z.); chenhaiping@sinopharm.com (H.C.); luolinyun@sinopharm.com (L.L.); liuxiaoqin6@sinopharm.com (X.L.); xiaoyanhui@sinopharm.com (Y.X.); yangyunkai@sinopharm.com (Y.Y.); 2Peking University Center for Public Health and Epidemic Preparedness & Response, Department of Epidemiology & Biostatistics, School of Public Health, Peking University, No. 38 Xueyuan Road, Haidian District, Beijing 100191, China; lvjun@bjmu.edu.cn (J.L.); han.yuting@pku.edu.cn (Y.H.); 3Department of Epidemiology & Biostatistics, School of Public Health, Peking University, Beijing 100191, China; 4Hunan Provincial Center for Disease Control and Prevention, Changsha 410005, China; ymlc01@hncdc.com (T.H.); ymlc06@hncdc.com (S.Z.); 5Guizhou Provincial Center for Disease Control and Prevention, Guiyang 550004, China; zrz19780816@163.com (R.Z.); muqiuyue831013@163.com (Q.M.); guanqh1977897@163.com (Q.G.); 6Fujian Provincial Center for Disease Control and Prevention, Fuzhou 350012, China; zhangdj@fjcdc.com.cn (D.Z.); huangrongdong@fjmu.edu.cn (R.H.); okxfq@163.com (F.X.); 7School of Public Health, Tongji Medical School, Huazhong University of Science and Technology, Wuhan 430074, China; shengwei@hust.edu.cn; 8Center for Evidence-Based and Translational Medicine, Zhongnan Hospital of Wuhan University, Wuhan 430071, China; huangjiao1019@163.com; 9Linli County Center for Disease Control and Prevention, Changde 415200, China; lljk_gh@163.com; 10Beijing Institute of Biological Products Co., Ltd., Beijing 100176, China; wh6247@126.com

**Keywords:** COVID-19 vaccine (Vero cell), inactivated, elderly hypertensive population, elderly diabetic population, safety, immunogenicity

## Abstract

Background: To evaluate the immunogenicity and safety of the COVID-19 vaccine (Vero cell), inactivated, in a population aged ≥60 years with hypertension or(/and) diabetes mellitus. Methods: A total of 1440 participants were enrolled and divided into four groups, 330 in the hypertension group, 330 in the diabetes group, 300 in the hypertensive combined with diabetes group (combined disease group), and 480 in the healthy population group. Two doses of the COVID-19 vaccine (Vero cell), inactivated, were administered at a 21-day interval and blood samples were collected before vaccination and 28 days after the second dose to evaluate the immunogenicity. The adverse events and changes in blood pressure and blood glucose levels after vaccination were recorded. Results: The seroconversion rate of the COVID-19 neutralizing antibodies was 100% for all participants. The post-inoculation geometric mean titer (GMT) in the four groups of the hypertension, diabetes, combined disease, and healthy populations were 73.41, 69.93, 73.84, and 74.86, respectively. The seroconversion rates and post-vaccination GMT in the hypertension, diabetes, and combined disease groups were non-inferior to the healthy population group. The rates of vaccine-related adverse reactions were 11.93%, 14.29%, 12.50%, and 9.38%, respectively. No serious adverse events were reported during the study. No apparent abnormal fluctuations in blood pressure and blood glucose values were observed after vaccination in participants with hypertension or(/and) diabetes. Conclusions: The COVID-19 vaccine (Vero cell), inactivated, showed good immunogenicity and safety in patients aged ≥60 years suffering from hypertension or(/and) diabetes mellitus.

## 1. Introduction

Since the outbreak of Coronavirus disease (COVID-19) in late 2019, a massive global pandemic ensued with an enormous public health crisis and devastating socioeconomic impact. To date, 526 million people have been infected, resulting in 6.28 million life losses [1]. Mass COVID-19 vaccination is the most effective prevention and control measure of the pandemic. As of 23 May 2022, 11.81 billion doses of vaccination have been administered worldwide [1]. Since 17 April 2022, the inactivated vaccine from Beijing Institute of Biological Products Co., Ltd. (Beijing, China), (Sinopharm BBIBP-CorV) has been included in the WHO list of COVID-19 vaccines for emergency use and over 1.2 billion doses have been administered domestically.

The world’s older population continues to grow at an unprecedented rate with an increasing prevalence of chronic diseases. Meanwhile, the number of people with multiple underlying conditions is also rising. Several studies have shown that advanced age and co-morbidities lead to higher rates of hospitalization and mortality after COVID-19 infection due to compromised immunity [2,3]. It is of paramount importance to implement large-scale vaccination against COVID-19 in this population to protect them from COVID-19.

Previous studies on specific groups, such as liver transplant recipients [4], patients with chronic kidney disease [5], patients with non-alcoholic fatty liver disease [6], and patients with hematopoietic stem cell transplantation [7], showed that the post-vaccination geometric mean titer (GMT) decreased variably compared with the healthy population. Compared with these diseases, cardiovascular and metabolic diseases, including hypertension and diabetes, are more prevalent in the elderly. In the United States, 74.5% of the elderly population has hypertension [8] and 26.8% of the elderly population has diabetes mellitus [9]. In China, nearly half (44.7%) of Chinese adults aged 35–75 years had hypertension [10] and 11.2% of adults aged ≥18 years had diabetes [11]. Among adults aged ≥60 years in China, the prevalence rates of hypertension increased rapidly from 1991 to 2015 (32.38% in urban areas and 30.46% in rural areas) [12]. Another study conducted in Haikou showed the prevalence of hypertension was 26.0% and that of diabetes mellitus was 8.0% in 2018 [13].

However, data on immunogenicity and safety in the healthy elderly population are limited in a few studies with small sample sizes [14,15,16]. The immunogenicity analysis and safety observation of the COVID-19 vaccines specifically for the elderly hypertensive and diabetic populations are yet to be reported. In addition, it is not clear whether COVID-19 vaccines have negative effects on blood pressure and blood glucose among elderly hypertensive and diabetic groups. 

In this study, we selected an elderly population aged ≥60 years with hypertension and diabetes and conducted an observational study on the immunogenicity and safety of inactivated COVID-19 vaccines. We evaluated the immunogenicity by measuring the seroconversion rate and post-vaccination GMT of SARS-CoV-2 antibodies before and after vaccination. The incidence of adverse events was monitored actively to estimate the safety of the vaccine. The blood pressure and blood glucose fluctuations were recorded to evaluate the possible short-term effects of vaccination on patients suffering from hypertension and diabetes.

## 2. Methods

### 2.1. Study Design and Participants

This observational study was sited in Hunan, Guizhou, and Fujian provinces of China. The study was approved by the Ethics Committee of Hunan Provincial Center for Disease Control and Prevention, Clinical Research Unit (approval number: Xiang CDC IRB-PJ2021005). A cohort of ≥60-year-olds with hypertension and diabetes were selected for the study between October 2021 and January 2022. Based on the results of phase I and II clinical studies, the seroconversion rate for people aged ≥60 years was 92% [14]. Using the serum antibody seroconversion rate as an index for immunogenicity validation assessment, the non-inferiority of the seroconversion rate in the test group compared with the control group was tested by taking a one-sided test level α of 0.025 and a test efficacy 1-β of 80%. The sample size estimation was a minimum of 178 study participants in each group with consideration of a 15% withdrawal rate; there were at least 210 study participants in each group (Beijing Institute of Biological Products Co., Ltd., Beijing, China).

The pre-vaccination body temperature of all participants should be less than 37.3 °C. Patients with hypertension/diabetes need to be diagnosed with hypertension and/or diabetes by a primary healthcare facility or higher. For hypertensive patients, the blood pressure on the day of vaccination should be <160 mmHg systolic and <100 mmHg diastolic through lifestyle modification and/or medication; for diabetic patients, the fasting blood glucose value on the day of vaccination (or within the previous three days) should be ≤13.9 mmol/L through lifestyle modification and/or medication. Exclusion criteria for participants include previous confirmed case of COVID-19 or asymptomatic infection; history of COVID-19 vaccination; known anaphylaxis to any component (including excipients) contained in the product; injection of non-specific immunoglobulin within 1 month prior to enrolment; live attenuated vaccine within 1 month prior to vaccination and other vaccines within 14 days prior to vaccination; previous severe allergic reaction to vaccine (e.g., acute allergic reaction, urticaria, angioneurotic edema, respiratory distress, etc.); and uncontrolled epilepsy and other progressive neurological disorders, history of Guillain–Barré syndrome, etc.

The specific grouping was 330 elderly hypertensive patients, 330 elderly diabetic patients, 300 elderly patients with combined hypertension and diabetes mellitus, and 480 healthy elderly population as controls. A total 70% of the population was aged 60–69 and 30% was aged 70 and over across the study groups. All participants were required to sign informed consent form before enrolled into each study group. The study was registered on clinicaltrials.gov (registration number: NCT05065879).

### 2.2. Study Vaccine

The COVID-19 vaccine (Vero cell), inactivated, was produced by the Beijing Institute of Biological Products Co., Ltd., based on the 19nCov-CDC-Tan-HB02 strain. It contains 6.5 U of the inactivated SARS-CoV-2 antigen in 0.5 mL (lot number 2021050799). The vaccination schedule is to administer two doses of inactivated COVID-19 vaccine, by intramuscular injection into the deltoid muscle of the upper arm, with the interval of 21 days.

### 2.3. Immunogenicity and Safety Evaluation

Blood samples of 5 mL were collected, and serum antibody levels were tested using a plaque reduction neutralization test at the National Institute for Viral Disease Control and Prevention, China CDC before the first dose of vaccine and 28 days after the second dose of vaccine. The detailed method for SARS-CoV-2 neutralizing assay was as follows: Serum samples were measured for neutralization capacity testing using infectious SARS-CoV-2 virus (strain 19nCoVCDC-Tan-Strain04 [QD01]) by the 50% cell culture infectious dose. Serum was successively diluted 1:4 to the required concentration by a 2-fold series, and an equal volume of challenge virus solution was added. After neutralization in a 37 °C incubator for 2 h, a 1.0~2.5 × 105/mL cell suspension was added to the wells (0.1 mL per well) and cultured in a CO2 incubator at 37 °C for 4 days. Titers expressed as the reciprocal of the highest dilution protecting 50% cell from virus challenge. Convalescent sera are included as an internal positive control in every assay. Serum testing technicians were masked to the specific study subgroups. Based on the test results, the seroconversion rate of SARS-CoV-2 neutralizing antibodies and the GMT level were evaluated. Seroconversion rate (seroconversion) was defined as post-injection titer of at least 1:16 if the baseline titer was below 1:4 or at least a 4-fold increase in post-vaccination titer from baseline if the baseline titer was at least 1:4.

Participants were observed for 30 min after each dose of vaccine and the occurrence of adverse events within 21 days of vaccination was recorded in a diary logbook. The solicited adverse events recorded in the diary included local adverse events (soreness, hardness, swelling, rash, redness, pruritus, erythema) and systemic adverse events (fever, dizziness, headache, cough, fatigue/lethargy, nausea, vomiting, chest tightness, diarrhea, constipation, dysphagia, anorexia, muscle pain (non-injection site), arthralgia, dyspnea, non-injection site pruritus (no skin lesions), skin mucosal abnormalities, acute allergic reactions, facial nerve symptoms). Serious adverse events (SAEs) were collected through telephone interviews and actively reporting by all participants. The classification of adverse reactions was in accordance with the *Guidelines for classification standards of Adverse Events in Clinical Trials of*
*Prophylactic vaccine**s issued by the State Medical Products Administration of China*. 

### 2.4. Blood Pressure and Blood Glucose Measurement

Blood pressure upper arm monitor (Brand: Omron) and blood glucose monitor (Brand: Verio Flex) were purchased by researchers. Each participant in the hypertension/combined disease group was given a blood pressure monitor, and each in the diabetes/combined disease group was given a blood glucose monitor. For those with hypertension, blood pressure was measured at 9 time points, before vaccination, 30 min after vaccination, and on days 1–7 after vaccination. Data before and 30 min after vaccination were acquired by staff on site, and others were tested at home by participants with the Omron blood pressure monitor. All data were recorded accurately in the diary logbook. In the diabetic population, fasting and 2 h postprandial blood glucose was measured at 5 time points, before and on days 1, 3, 5, and 7 after vaccination. All measurements were conducted by participants with the Verio Flex blood glucose monitor and data were recorded accurately in the diary logbook as well. Specialized staff trained hypertension/diabetes/combined disease participants on site on how to use blood pressure and blood glucose monitors to ensure that they were operated and recorded accurately and met the requirements of specific measurement time points.

### 2.5. Statistical Analysis

We report baseline characteristics using descriptive statistics and summarize continuous variables using mean, standard deviation, median, and categorical variables using n (%). Immunogenicity was statistically analyzed using SAS 9.4 statistical software. Differences in rates between groups were compared using the Chi-square test or Fisher’s exact test; differences in antibody GMT between groups were compared using the generalized linear model (GLM) and differences in antibody GMT between the three disease groups and healthy controls were compared using multiple comparisons (DUNNETT test). We further stratified the analyses by ages and genders. A non-inferiority test was performed on the post-vaccination antibody seroconversion rate, the lower bound of over −10% of the 95% confidence interval of the difference between the post-vaccination antibody seroconversion rate of the hypertension group/diabetes group/combined disease group and the healthy control group was used as the criterion for determining non-inferiority. For the non-inferiority test of post-vaccination antibody GMT, the lower bound of over 0.67 of the 95% confidence interval of the post-vaccination antibody GMT ratio between the hypertension group/diabetes group/combined disease group and the healthy control group was used as the criterion for non-inferiority. The mean values of blood pressure and blood glucose of each group and at various time points were used to describe the changes in blood pressure and blood glucose before and after vaccination.

### 2.6. Patient and Public Involvement

Although study participants contributed in important ways to this study, it was not feasible to involve them in the design, conduct, reporting, or dissemination plans of our study. We did not involve members of the public in this research owing to resource and time constraints.

## 3. Results

### 3.1. Study Population

A total of 1440 study participants were enrolled in this study. With the exclusion of 8 cases of protocol violations due to an inclusion error, a total of 1432 study participants (327 in the hypertension group, 329 in the diabetes group, 296 in the comorbid disease group, and 480 in the healthy control group) were included in the safety analysis data set. With the exclusion of 19 cases who failed to obtain blood samples after vaccination or missed the window for the collection time, a total of 1413 study participants (325 in the hypertension group, 328 in the diabetes group, 292 in the combined disease group, and 468 in the healthy controls) were included in the immunogenicity analysis data set. The flow chart of the study participants’ grouping and inclusion are shown in Figure 1, and the specific number distribution is shown in Table 1.

### 3.2. Immunogenicity of the Vaccine

The neutralizing antibody testing results showed that all participants were negative for pre-vaccination antibodies and the post-vaccination neutralizing antibody seroconversion rate was 100%, and the seroconversion rate in the diseased group was non-inferior to that in the healthy population control group. The post-vaccination neutralizing antibody GMT in the hypertension, diabetes, combined disease, and healthy control group was 73.41 (95% CI, 67.24–80.14), 69.93 (95% CI, 63.59–76.89), 73.84 (95% CI, 66.70–81.75), and 74.86 (95% CI, 69.49–80.64), respectively. The difference in the neutralizing antibody GMT post vaccination was not statistically significant between the groups (*p* = 0.722). There were no significant difference for the GMTs among the four populations in males and females (*p* = 0.139, *p* = 0.964, respectively). The GMT ratio of the hypertension group, diabetic group, and combined disease group was 0.98 (95% CI, 0.87–1.10), 0.93 (95% CI, 0.83–1.05), and 0.99 (95% CI, 0.87–1.12), respectively, compared to the healthy control group. The lower bound of the 95% confidence interval for all ratios was greater than 0.67, indicating that the GMT levels in each group were all non-inferior to those of the healthy controls. After being adjusted for age and gender, there was no significant difference in the GMTs between the three disease populations and the healthy population (Appendix A).

The analysis of the immunogenicity results in the 60–69 age group showed that the post-vaccination neutralizing antibody GMT in the hypertension, diabetes, combined disease, and healthy control group was 74.05 (95% CI, 66.95–81.89), 72.93 (95% CI, 65.27–81.50), 78.81 (95% CI, 69.58–89.25), and 75.88 (95% CI, 69.43–82.93), respectively. The difference in the neutralizing antibody GMT post vaccination among groups was not statistically significant (*p* = 0.786). There was no significant difference for the GMTs among the four populations in males and females (*p* = 0.497, *p* = 0.744, respectively). The differences in the GMT levels were not statistically significant in the hypertension group (*p* = 0.977), the diabetes group (*p* = 0.911), and the comorbid disease group (*p* = 0.930) compared with the healthy control group.

The analysis of the immunogenicity results in the group aged 70 years and older showed the post-vaccination neutralizing antibody GMT in the hypertension, diabetes, combined disease, and healthy control group was 71.97 (95% CI, 60.32–85.86), 63.16 (95% CI, 52.53–75.94), 63.96 (95% CI, 53.63–76.27), and 72.56 (95% CI, 63.22–83.29), respectively. There was no statistically significant difference in the neutralizing antibody GMT post vaccination among groups (*p* = 0.500). There was no significant difference for the GMTs among the four populations in males and females (*p* = 0.204, *p* = 0.831, respectively). The differences in the GMT levels were not statistically significant in the hypertension group (*p* = 1.000), the diabetes group (*p* = 0.497), and the comorbid disease group (*p* = 0.585) compared with the healthy control group (Figure 2).

### 3.3. Safety of the Vaccine

The incidence of all adverse reactions within 21 days after two doses of the vaccine in the hypertension, diabetes group, combined disease group, and healthy control group was 11.93%, 14.29%, 12.50%, and 9.38% respectively. The difference in incidence between groups was not statistically significant (*p* = 0.185) The incidence of solicited adverse reactions was 10.70%, 13.07%, 10.14%, and 7.71%, respectively, with no statistical significance between groups (*p* = 0.096) (Table 2). However, the incidence of AEs was different between males and females (Appendix A), the 60–69-years-old group, and the 70 years and older group (Appendix A).

The incidence of localized adverse reactions was 3.98%, 5.47%, 4.05%, and 4.17%, respectively, in the hypertension group, diabetes group, combined disease group, and healthy control group within 21 days after two doses of the vaccine with no statistical significance between groups (*p* = 0.757); the symptoms were mainly soreness and swelling. Meanwhile, the incidence of systemic adverse reactions was 8.56%, 10.64%, 7.77%, and 5.42%, respectively, with no statistical significance between groups (*p* = 0.052); the symptoms were mainly dizziness, fatigue/lethargy, and headache. The severity of adverse reactions was all grade 1 (mild) mainly (Table 3). However, the severity of AEs was different between males and females (Appendix A), the 60–69-years-old group, and the 70 years and older group (Appendix A).

### 3.4. Analysis of Blood Pressure and Blood Glucose Levels

For those with hypertension, blood pressure was measured at nine time points, before vaccination, 30 min after vaccination, and on days 1–7 after vaccination, and the line graph did not show any significant abnormal fluctuations. In the diabetic population, fasting and 2 h postprandial blood glucose was measured at five time points, before and on days 1, 3, 5, and 7 after vaccination, and the line graph did not show any significant abnormal fluctuations (Figure 3).

## 4. Discussion

Our study results showed that the COVID-19 vaccine (Vero cell), inactivated, has good immunogenicity and safety in the population aged ≥60 years with hypertension and diabetes mellitus. The seroconversion rates and post-vaccination GMT in the hypertension, diabetes, and combined disease groups were non-inferior compared with the healthy controls. The differences in the seroconversion rates, post-vaccination GMT, and the incidence of adverse reactions among the four groups were not statistically significant. No serious adverse effects were reported during the study. No apparent abnormal fluctuations in blood pressure and blood glucose values were observed after vaccination in participants with hypertension and diabetes. 

The results of the study also showed that hypertension, diabetes, and combined illnesses did not adversely affect the immune response in the population aged ≥60 years. During the phase Ⅰ/Ⅱ clinical studies of Sinopharm BBIBP-CorV, 24 healthy subjects aged over 60 were included, with a post-vaccination GMT result of 18.9 (95% CI: 13.4–26.6) 28 days after the second dose of vaccine (6.5 U of antigen), relatively lower compared with our result (73.84). Another phase Ⅰ/Ⅱ clinical study of the inactivated COVID-19 vaccine among a healthy elderly population showed a plunge of the GMT (less than 8) six months after two doses of vaccination. The GMT increased rapidly to 342.8 after a booster inoculation [17]. The phase Ⅰ clinical study of this vaccine enrolled 96 younger subjects aged 18–59 with a neutralizing antibody GMT of 211.2 (95% CI: 158.9–280.6) on day 42 after the second inoculation. The phase 2 study enrolled 448 participants aged 18–59 years, and the vaccine-elicited neutralizing antibody titer on day 28 after the second inoculation was 218.0 (95% CI: 158.9–280.6). These results showed a higher post-vaccination GMT compared with our elderly cohort. We also find other studies showing that post-vaccination neutralizing antibody levels are weaker in the elderly population than in the younger population [18,19], including the BNT162b1 RNA vaccine and the BNT162b2 RNA vaccine [16]. 

Because elderly people, especially those with multiple co-morbidities such as hypertension and diabetes, are more susceptible to SARS-CoV-2 infection with a higher likelihood of developing severe disease after infection and a higher hospitalization rate and mortality rate [20], it is essential to encourage timely booster vaccination and lessen hesitancy [21]. Several previous studies have also provided similar advice [18,22].Therefore, the WHO and many countries (including China) recommend a booster shots program to be administered after 6 months of completion of the full primary series [23,24,25]. 

The active monitoring of all participants in this study within 21 days after vaccination revealed only dominated systemic reactions, including dizziness, fatigue/lethargy, and headache, all of which resolved, with a grade 1 (mild) severity. No vaccine-related serious adverse events were reported during the study, showing a good safety of the Sinopharm BBIBP-CorV vaccine. It has also shown good immunogenicity and safety in phase III clinical studies in the United Arab Emirates, Bahrain, Egypt, and Jordan [26]. The results showed the two most common adverse reactions were local injection site soreness and headache, which were predominantly mild and spontaneously subsided, similar to the results of this study. It is worth noting that some rare adverse reactions have emerged since the large-scale use of the COVID-19 vaccines, such as vaccine-induced immune thrombotic thrombocytopenia, cerebral venous sinus thrombosis [27,28], and some neurological side effects, including Guillain–Barre syndrome, transverse myelitis, Bell’s palsy, etc. [29,30,31,32]. Given the very low incidence of these serious adverse reactions, the beneficial and protective effects of the COVID-19 inactivated vaccine far outweigh the risks. It is reassuring to observe that active monitoring of all subjects in this study within 21 days after vaccination did not reveal any of the above-mentioned adverse events.

Our study is the first ever to report the immunogenicity and safety in a hypertensive and diabetic population aged ≥60 years after COVID-19 inactivated vaccine, as well as the possible short-term influence on their blood pressure and blood glucose. Thus far, there have been no exploratory studies on the effects of COVID-19 vaccination on blood pressure and blood glucose. In this study, we minimized the measurement error by providing the same model of blood glucose and blood pressure monitor to the participants and training them in a standardized way. Blood pressure and blood glucose testing at multiple time points before and after vaccination showed no significant impact among elderly people with hypertension and diabetes. 

We plan to follow up with these participants with different immunization schedules of booster shots to obtain immunogenicity results to further carry out antibody persistence studies in the hope of providing a more thorough evidence-based basis for further improvement of the immunization schedule of the COVID-19 inactivated vaccine.

## 5. Limitations

This study has several limitations. Due to ethical reasons, we did not set up unvaccinated or placebo-vaccinated healthy control groups. Because the study period was right amidst the COVID-19 pandemic, it would have been unethical to leave a portion of the subjects exposed to the threat without vaccination, especially the elderly population with underlying illnesses. In addition, given that most of the healthy population had already been vaccinated against COVID-19 at the time, the number of healthy population controls that could be enrolled was not too large. Another limitation of the study is that antibody duration data after vaccination in this population are still lacking, as well as the lack of long-term blood pressure and glucose results among the hypertension and diabetes population.

## 6. Conclusions

With a 100% seroconversion rate of the COVID-19 neutralizing antibodies for all participants, the vaccine has showed good immunogenicity in the population aged ≥60 years suffering from hypertension and diabetes mellitus. No serious adverse events were reported during the study, showing the vaccine has satisfying safety. In the future, further studies with larger sample sizes and longer research duration are still necessary.

## Figures and Tables

**Figure 1 vaccines-10-01020-f001:**
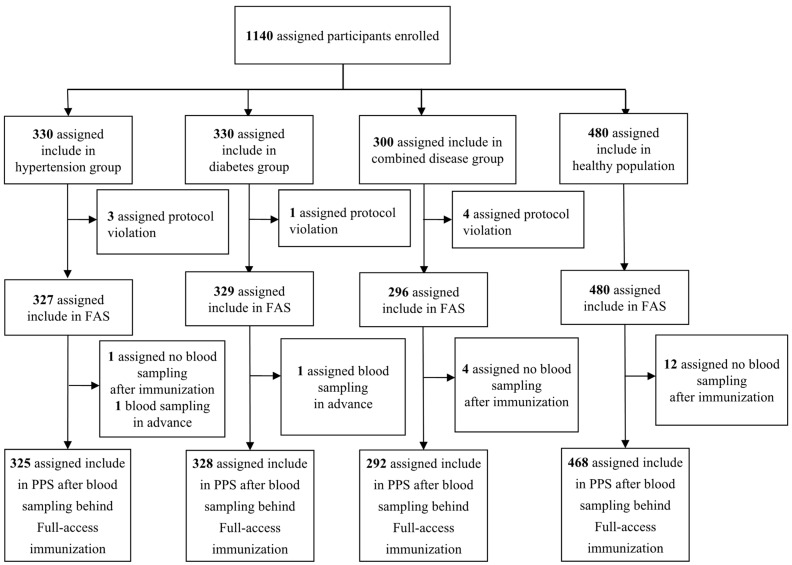
Screening and Enrollment of Participants. A total of 1440 study participants were enrolled in this study. A total of 1432 study participants were included in the safety analysis data set. A total of 1413 study participants were included in the immunogenicity analysis data set.

**Figure 2 vaccines-10-01020-f002:**
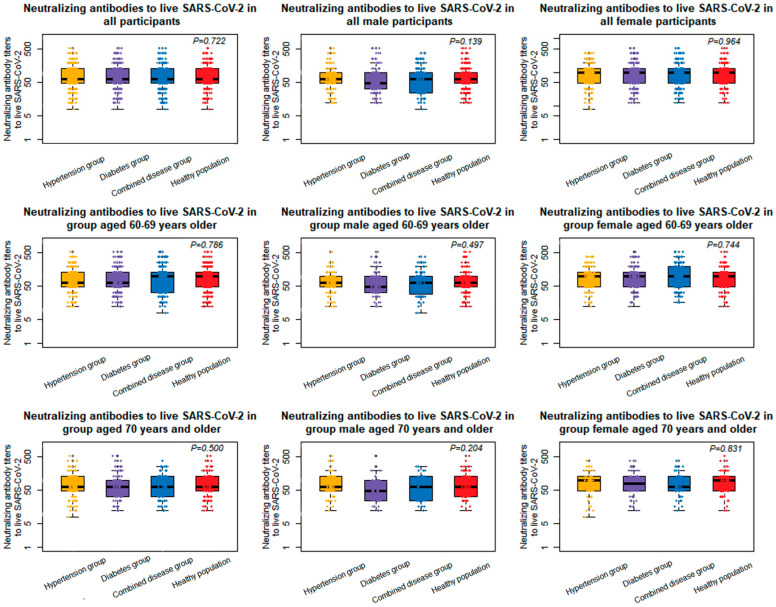
The level of neutralizing antibodies to live SARS-CoV-2 in adults aged over 60 years. Dots are reciprocal neutralizing antibody titers for individuals in the per-protocol population. Only *p* values of comparisons among four groups were shown (all *p* values are greater than 0.05).

**Figure 3 vaccines-10-01020-f003:**
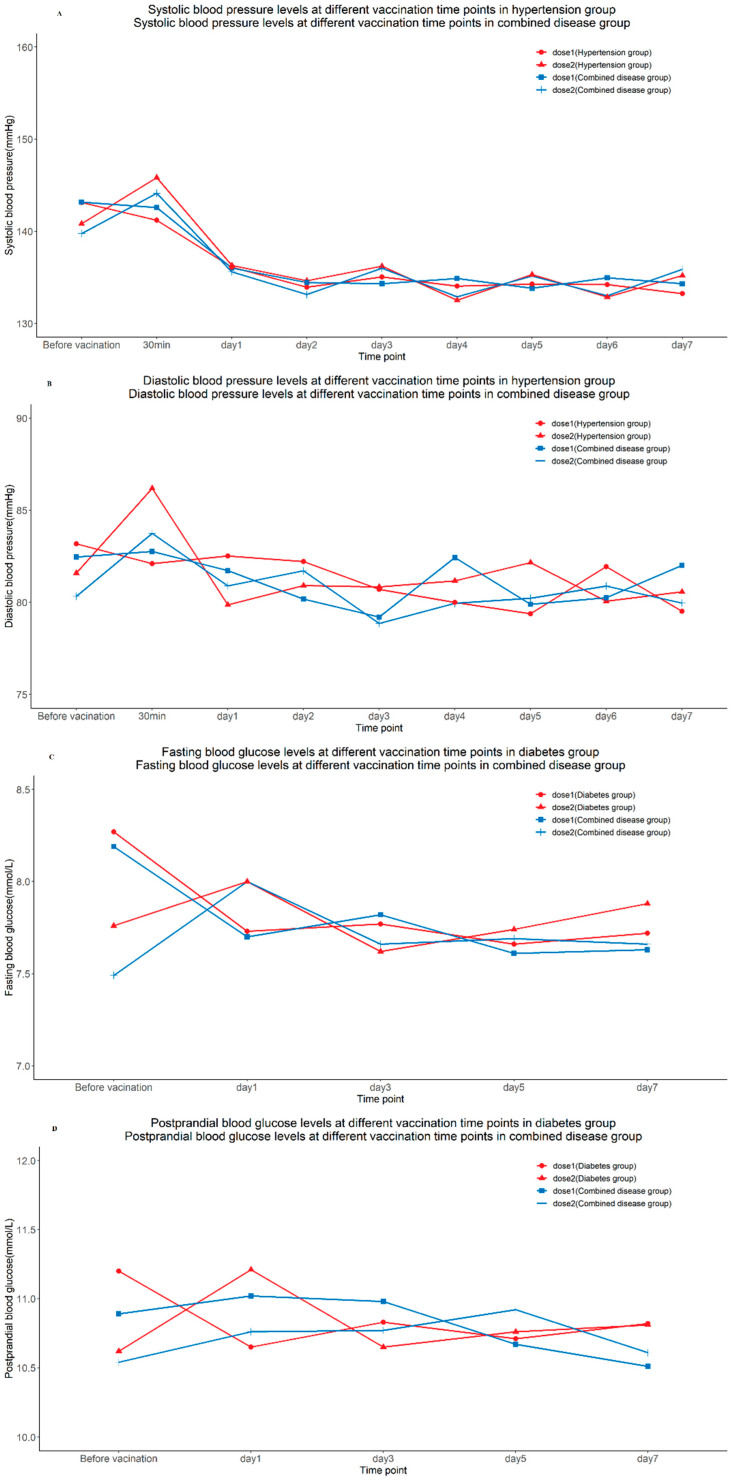
The blood pressures and blood glucose levels at different vaccination time points. (**A**) Systolic blood pressure levels at different vaccination time points in hypertension group and combined disease group. (**B**) Diastolic blood pressure levels at different vaccination time points in hypertension group and combined disease group. (**C**) Fasting blood glucose levels at different vaccination time points in diabetes group and postprandial blood glucose levels at different vaccination time points in combined disease group. (**D**) Postprandial blood glucose levels at different vaccination time points in diabetes group and combined disease group.

**Table 1 vaccines-10-01020-t001:** Baseline demographic characteristics in the immunogenicity analysis data set (n = 1413).

Characteristics	Hypertension Group	Diabetes Group	Combined Disease Group	Healthy Population	*p*
Age					0.279 *
Mean ± SD	68.79 ± 5.34	68.02 ± 4.62	68.33 ± 5.42	68.3 ± 5.09	
Median	68.18	67.42	67.55	67.4	
Min, Max	60.01, 96.46	60.31, 83.36	60.07, 96.4	60.11, 87.31	
Gender					0.001 **
Male	157 (48.31%)	137 (41.77%)	118 (40.41%)	249 (53.21%)	
Female	168 (51.69%)	191 (58.23%)	174 (59.59%)	219 (46.79%)	

* Analysis of Variance, ** Chi-square test.

**Table 2 vaccines-10-01020-t002:** Adverse events related to the study vaccine 0–21 days after two doses of vaccine in the whole population.

	Hypertension Group (327 Assigned)			Diabetes Group (329 Assigned)			Combined Disease Group (296 Assigned)			Healthy Population (480 Assigned)					
Adverse Events	Case Times	No. of Cases	Incidence Rate	Case Times	No. of Cases	Incidence Rate	Case Times	No. of Cases	Incidence Rate	Case Times	No. of Cases	Incidence Rate	Statistical Method	Statistic	*p*
Total	59	39	11.93	92	47	14.29	80	37	12.50	80	45	9.38	Chi-Square Test	4.828	0.1849
Solicited event	43	35	10.70	76	43	13.07	62	30	10.14	64	37	7.71	Chi-Square Test	6.336	0.0964
Local adverse events	13	13	3.98	21	18	5.47	15	12	4.05	24	20	4.17	Chi-Square Test	1.184	0.7569
Soreness	11	11	3.36	15	13	3.95	9	8	2.70	23	19	3.96	Chi-Square Test	1.038	0.7919
Swelling	0	0	0.00	5	5	1.52	3	2	0.68	0	0	0.00	Fisher’s Exact Test	-	0.0044
Pruritus	1	1	0.31	1	1	0.30	2	2	0.68	1	1	0.21	Fisher’s Exact Test	-	0.8262
Rash	1	1	0.31	0	0	0.00	0	0	0.00	0	0	0.00	Fisher’s Exact Test	-	0.4351
Hardness	0	0	0.00	0	0	0.00	1	1	0.34	0	0	0.00	Fisher’s Exact Test	-	0.2067
Systemic adverse events	30	24	7.34	55	31	9.42	47	21	7.09	40	21	4.38	Chi-Square Test	8.243	0.0413
Dizziness	14	12	3.67	23	17	5.17	8	7	2.36	13	10	2.08	Chi-Square Test	6.915	0.0746
Fatigue/lethargy	7	7	2.14	9	9	2.74	10	8	2.70	8	8	1.67	Chi-Square Test	1.397	0.7063
Headache	1	1	0.31	7	7	2.13	7	6	2.03	3	3	0.63	Fisher’s Exact Test	-	0.0474
Chest tightness	1	1	0.31	5	4	1.22	4	4	1.35	1	1	0.21	Fisher’s Exact Test	-	0.1187
Nausea	1	1	0.31	3	3	0.91	4	4	1.35	1	1	0.21	Fisher’s Exact Test	-	0.1607
Non-injection site pruritus	1	1	0.31	1	1	0.30	2	2	0.68	3	2	0.42	Fisher’s Exact Test	-	0.8498
Muscle pain	2	2	0.61	1	1	0.30	0	0	0.00	4	3	0.63	Fisher’s Exact Test	-	0.6524
Cough	0	0	0.00	2	2	0.61	3	3	1.01	2	2	0.42	Fisher’s Exact Test	-	0.2892
Arthralgia	1	1	0.31	0	0	0.00	3	2	0.68	1	1	0.21	Fisher’s Exact Test	-	0.4899
Fever	1	1	0.31	0	0	0.00	3	3	1.01	0	0	0.00	Fisher’s Exact Test	-	0.0152
Diarrhoea	0	0	0.00	1	1	0.30	2	2	0.68	0	0	0.00	Fisher’s Exact Test	-	0.0913
Anorexia	0	0	0.00	1	1	0.30	0	0	0.00	2	2	0.42	Fisher’s Exact Test	-	0.7034
Constipation	0	0	0.00	0	0	0.00	1	1	0.34	1	1	0.21	Fisher’s Exact Test	-	0.6927
Vomiting	1	1	0.31	0	0	0.00	0	0	0.00	1	1	0.21	Fisher’s Exact Test	-	0.8459
Skin mucosal abnormalities	0	0	0.00	2	2	0.61	0	0	0.00	0	0	0.00	Fisher’s Exact Test	-	0.1473

**Table 3 vaccines-10-01020-t003:** Severity of adverse events 0–21 days after two doses of vaccine in the whole population.

	Hypertension Group (327 Assigned)			Diabetes Group (329 Assigned)			Combined Disease Group (296 Assigned)			Healthy Population (480 Assigned)			
Adverse Events	Level 1	Level 2	Level 3	Level 1	Level 2	Level 3	Level 1	Level 2	Level 3	Level 1	Level 2	Level 3	KW_*p*
Total	45 (13.76)	9 (2.75)	6 (1.83)	46 (13.98)	18 (5.47)	4 (1.22)	31 (10.47)	16 (5.41)	6 (2.03)	53 (11.04)	23 (4.79)	5 (1.04)	0.6145
Solicitation event	36 (11.01)	2 (0.61)	1 (0.31)	38 (11.55)	8 (2.43)	1 (0.30)	22 (7.43)	7 (2.36)	3 (1.01)	37 (7.71)	4 (0.83)	0 (0.00)	0.0728
Local adverse events	13 (3.98)	0 (0.00)	0 (0.00)	18(5.47)	1 (0.30)	0 (0.00)	11 (3.72)	2 (0.68)	0 (0.00)	20 (4.17)	0 (0.00)	0 (0.00)	0.6638
Soreness	11 (3.36)	0 (0.00)	0 (0.00)	13(3.95)	0 (0.00)	0 (0.00)	7 (2.36)	1 (0.34)	0 (0.00)	19 (3.96)	0 (0.00)	0 (0.00)	0.7968
Swelling	0 (0.00)	0 (0.00)	0 (0.00)	4(1.22)	1 (0.30)	0 (0.00)	1 (0.34)	1 (0.34)	0 (0.00)	0 (0.00)	0 (0.00)	0 (0.00)	0.0100
Pruritus	1 (0.31)	0 (0.00)	0 (0.00)	1(0.30)	0 (0.00)	0 (0.00)	2 (0.68)	0 (0.00)	0 (0.00)	1 (0.21)	0 (0.00)	0 (0.00)	0.7490
Rash	1 (0.31)	0 (0.00)	0 (0.00)	0 (0.00)	0 (0.00)	0 (0.00)	0 (0.00)	0 (0.00)	0 (0.00)	0 (0.00)	0 (0.00)	0 (0.00)	0.3368
Hardness	0 (0.00)	0 (0.00)	0 (0.00)	0 (0.00)	0 (0.00)	0 (0.00)	1 (0.34)	0 (0.00)	0 (0.00)	0 (0.00)	0 (0.00)	0 (0.00)	0.2795
Systemic adverse events	28 (8.56)	4 (1.22)	1 (0.31)	41 (12.46)	10 (3.04)	3 (0.91)	34 (11.49)	6 (2.03)	8 (2.70)	35 (7.29)	5 (1.04)	0 (0.00)	0.0003
Dizziness	12 (3.67)	1 (0.31)	0 (0.00)	17 (5.17)	2 (0.61)	0 (0.00)	6 (2.03)	1 (0.34)	0 (0.00)	12 (2.50)	0 (0.00)	0 (0.00)	0.0528
Fatigue/lethargy	7 (2.14)	0 (0.00)	0 (0.00)	7 (2.13)	1 (0.30)	1 (0.30)	5 (1.69)	1 (0.34)	2 (0.68)	8 (1.67)	1 (0.21)	0 (0.00)	0.8160
Headache	3 (0.92)	0 (0.00)	0 (0.00)	4 (1.22)	2 (0.61)	1 (0.30)	6 (2.03)	0 (0.00)	1 (0.34)	3 (0.63)	0 (0.00)	0 (0.00)	0.1146
Cough	1 (0.31)	1 (0.31)	0 (0.00)	1 (0.30)	1 (0.30)	0 (0.00)	3 (1.01)	1 (0.34)	1 (0.34)	1 (0.21)	1 (0.21)	0 (0.00)	0.2323
Nausea	0 (0.00)	1 (0.31)	0 (0.00)	3 (0.91)	0 (0.00)	1 (0.30)	4 (1.35)	0 (0.00)	0 (0.00)	1 (0.21)	0 (0.00)	0 (0.00)	0.1408
Chest tightness,	1 (0.31)	0 (.00)	0 (0.00)	4 (1.22)	0 (0.00)	0 (0.00)	4 (1.35)	0 (0.00)	0 (0.00)	1 (0.21)	0 (0.00)	0 (0.00)	0.1401
Muscle pain	1 (0.31)	0 (0.00)	1 (0.31)	2 (0.61)	0 (0.00)	0 (0.00)	0 (0.00)	0 (0.00)	0 (0.00)	4 (0.83)	0 (0.00)	0 (0.00)	0.5046
Non-injection site pruritus	1 (0.31)	0 (0.00)	0 (0.00)	1 (0.30)	0 (0.00)	0 (0.00)	2 (0.68)	1 (0.34)	0 (0.00)	1 (0.21)	1 (0.21)	0 (0.00)	0.5350
Arthralgia	0 (0.00)	1 (0.31)	0 (0.00)	0 (0.00)	1 (0.30)	0 (0.00)	2 (0.68)	1 (0.34)	0 (0.00)	1 (0.21)	1 (0.21)	0 (0.00)	0.5377
Fever	1 (0.31)	0 (0.00)	0 (0.00)	0 (0.00)	0 (0.00)	0 (0.00)	1 (0.34)	0 (0.00)	2 (0.68)	0 (0.00)	0 (0.00)	0 (0.00)	0.0459
Diarrhea	0 (0.00)	0 (0.00)	0 (0.00)	0 (0.00)	1 (0.30)	0 (0.00)	0 (0.00)	1 (0.34)	2 (0.68)	0 (0.00)	0 (0.00)	0 (0.00)	0.0461
Skin mucosal abnormalities	0 (0.00)	0 (0.00)	0 (0.00)	2 (0.61)	1 (0.30)	0 (0.00)	0 (0.00)	0 (0.00)	0 (0.00)	0 (0.00)	0 (0.00)	0 (0.00)	0.0180
Anorexia	0 (0.00)	0 (0.00)	0 (0.00)	0 (0.00)	1 (0.30)	0 (0.00)	0 (0.00)	0 (0.00)	0 (0.00)	2 (0.42)	0 (0.00)	0 (0.00)	0.4880
Constipation	0 (0.00)	0 (0.00)	0 (0.00)	0 (0.00)	0 (0.00)	0 (0.00)	1 (0.34)	0 (0.00)	0 (0.00)	0 (0.00)	1 (0.21)	0 (0.00)	0.5912
Vomiting	1 (0.31)	0 (0.00)	0 (0.00)	0 (0.00)	0 (0.00)	0 (0.00)	0 (0.00)	0 (0.00)	0 (0.00)	1 (0.21)	0 (0.00)	0 (0.00)	0.6408

When the same subject has the same adverse reaction several times, the most serious one shall be taken.

## Data Availability

Data are available for scientific purposes after written request to the corresponding author.

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
