# Peer review of "Evaluation of Immunogenicity and Safety of Vero Cell-Derived Inactivated COVID-19 Vaccine in Older Patients with Hypertension and Diabetes Mellitus"

_vaccines, 2022, doi:10.3390/vaccines10071020_

Round 1

Reviewer 1 Report

The authors conducted a very basic study on subjects with diabetes and hypertension. However, the study does have some scientific merit, given the significant number of participants and their anxieties about taking covid shots. 

Additional comments: 

Overall, the manuscript is fairly well written. However, there are several grammatical mistakes like missing punctuation marks, typos, and poor sentence structure in many manuscript sections. Avoid repetitious text throughout the manuscript. 

Poor referencing throughout the text. Authors should cite more updated references to support the text. 

The figures need to be a shaper. 

Author Response

Point 1: Overall, the manuscript is fairly well written. However, there are several grammatical mistakes like missing punctuation marks, typos, and poor sentence structure in many manuscript sections. Avoid repetitious text throughout the manuscript.

Response 1: Thank you for your suggestion. We have found a professional institution to polish the language in the paper. Hope this revised version could make the paper more readable.

Point 2: Poor referencing throughout the text. Authors should cite more updated references to support the text.
Response 2: We have updated WHO's documents and added some updatde references as the reviewer’s suggestion.

Point 3: The figures need to be a shaper.
Response 3: As your suggestion, we have change the figures to 300DPI and be sharper than the previous version.

Reviewer 2 Report

The objective of the current study was the immunogenicity and safety of inactivated COVID-19 vaccines e (Vero cell) inactivated in population aged ≥ 60 years with hypertension or/and diabetes mellitus. The manuscript includes good results however some major concerns were raised.

Major comments

L70-71: it is better to show the prevalence of these disease in the elderly or those with age of > 60 years (the studied group in the current study).

L80: why authors selected this group (60 years) since the elderly definition those people aged 50 years or more

Update the epidemiological numbers in the introduction

Table 2: why authors used different statistical methods (Chi-Square and Fisher's Exact Tests)?

The discussion section should be expanded and further studies about the immunogenicity, safety and side effects of COVID-19 vaccines should be added. The results should be correlated with the characteristics the type of vaccine in this study and compare with other vaccine types.   

The conclusion section contained only the authors' plan for further studies so it is not well written, and it should be modified to show the general conclusions of the current study and then add recommendation or further studies.

Figures 2 and 3 are not clear.

Minor comments

L49-57: update the numbers in this paragraph.

L:59: Meanwhile, the

L66: stem cell transplantation7 showed

L66: "GMT" write for the first time in full words.

L83: replace "geometric mean titre (GMT)" with GMT

L93: IRB-PJ2021005). A cohort of

Replace "aged 60 years and above" with "≥ years" throughout the manuscript.

L131: blood samples of 5ml were

Reviewer 3 Report

In this manuscript, authors described the immunogenicity and safety profiles of inactivated COVID-19 vaccine (Vero cell) in elderly population of age above 60 years. This study is important as it explores antibody production and adverse reactions among elderly with comorbidities like hypertension and diabetes. However, there are multiple aspects to improve and clarify for this manuscript:

1. Gender differences is observed in terms of immunogenicity and adverse reactions development during vaccination. As the authors have shown significant differences in terms of gender of participants in different groups, it is important that they analyze and present data as the interaction of age and sex. 

2. Data presentation and discussion should take care of the fact that they do not have unvaccinated or placebo-vaccinated healthy control group in this study. The observation of no increased adverse events is based on comparison with vaccinated healthy controls. Therefore, this fact (not having placebo control group) is a big limitation and needs to be clarified throughout the manuscript. In discussion, authors need to expand how immunogenicity and adverse events of this vaccine in elderly (this study) compared with that in younger adults (earlier studies).

3. Looks like combined disease group has more evidence of adverse reactions compared to other groups as there are significant differences in different local and systemic adverse events. This needs to be better explained. Also, the purpose of table 3 is not clear and this has never been explained in results/discussion sections. 

Minor:

1. Make sure whether it is geometric mean or geometric median titer. 

2. Mention how neutralizing antibodies were determined. 

3. Improve the quality of figure 3. 

4. 'COVID-19 virus' is wrong and should be replaced with 'SARS-CoV-2' throughout the manuscript. 

Round 2

Reviewer 2 Report

The manuscript has been sufficiently improved and can be published in the current form. 

Author Response

Thanks for your acknowledgment.

Reviewer 3 Report

Previous comments are addressed. Still, following changes are suggested:

1. Change title as 'Evaluation of immunogenicity and safety of COVID-19 vaccine (vero cell) inactivated in older patients with hypertension and diabetes mellitus.'

2. Include neutralizing antibody method in main paper instead of appendix. 

3. Appendix table 1 shows gender differences in antibody response but it is not stated in result. 

4. In Figure 2, remove the p values as they are above 0.05 (non-significant). It is understood that there is no significant difference. 
